# AMATA: AN ANNEALING MECHANISM FOR ADVERSARIAL TRAINING ACCELERATION

## ABSTRACT

Despite of the empirical success in various domains, it has been revealed that deep neural networks are vulnerable to maliciously perturbed input data that much degrade their performance. This is known as adversarial attacks. To counter adversarial attacks, adversarial training formulated as a form of robust optimization has been demonstrated to be effective. However, conducting adversarial training brings much computational overhead compared with standard training. In order to reduce the computational cost, we propose a simple yet effective modification to the commonly used projected gradient descent (PGD) adversarial training by increasing the number of adversarial training steps and decreasing the adversarial training step size gradually as training proceeds. We analyze the optimality of this annealing mechanism through the lens of optimal control theory, and we also prove the convergence of our proposed algorithm. Numerical experiments on standard datasets, such as MNIST and CIFAR10, show that our method can achieve similar or even better robustness with around 1/3 to 1/2 computation time compared with PGD.

## 1 INTRODUCTION

Recently, the revival of deep neural networks has led to breakthroughs in various fields, including computer vision, natural language processing, game playing, etc. Despite these advancements, deep neural networks were found to be vulnerable to malicious perturbations on the original input data. While the perturbations remain almost imperceptible to humans, they can lead to wrong predictions over the perturbed examples (Szegedy et al., 2013; Goodfellow et al., 2014; Akhtar & Mian, 2018). These maliciously crafted examples are known as adversarial examples, which have caused serious concerns over the reliability and security of deep learning systems, particularly when deployed in the life-critical scenarios, such as autonomous driving systems and health/medical domains.

Several defense mechanisms have been proposed, such as input reconstruction (Meng & Chen, 2017; Song et al., 2018), input encoding (Buckman et al., 2018), and adversarial training (Goodfellow et al., 2014; Tramèr et al., 2017; He et al., 2017; Madry et al., 2017). Among these methods, adversarial training is the most effective defense method so far. Adversarial training can be posed as a robust optimization problem (Ben-Tal & Nemirovski, 1998), where a min-max optimization problem is solved (Madry et al., 2017; Kolter & Wong, 2017). For example, given a $C$-class dataset $S = \{(\mathbf{x}_i^0, y_i)\}_{i=1}^n$ with $\mathbf{x}_i^0 \in \mathcal{R}^d$ as a normal or clean example in the $d$-dimensional input space and $y_i \in \mathcal{R}^C$ as its associated one-hot label, the objective of adversarial training is to solve the following *min-max optimization* problem:

$$\min_{\boldsymbol{\theta}} \frac{1}{N} \sum_{i=1}^{N} \max_{\|\mathbf{z}_i - \mathbf{x}_i^0\| \leq \epsilon} \ell(h_{\boldsymbol{\theta}}(\mathbf{z}_i), y_i) \tag{1}$$

where $h_{\boldsymbol{\theta}} : \mathcal{R}^d \to \mathcal{R}^C$ is the deep neural network (DNN) function, $\ell$ is the loss function and $\epsilon$ is the maximum perturbation constraint. The *inner maximization* problem is to find an adversarial example $\mathbf{x}_i$, within the $\epsilon$-ball around a given normal example $\mathbf{x}_i^0$ that maximizes the classification loss $\ell$. On the other hand, the *outer minimization* problem is to find model parameters that minimizes the loss $\ell$ on the adversarial examples $\{\mathbf{x}_i\}_{i=1}^n$ that are generated from the inner maximization.

The inner maximization problem is commonly solved by projected gradient descent (PGD). PGD perturbs a normal example $\mathbf{x}^0$ by iteratively updating it in the steepest ascent direction for a total

of $K$ times. Each ascent step is modulated by a small step size and a projection step back onto the $\epsilon$-ball of $\mathbf{x}^0$ to prevent the updated value fall outside the $\epsilon$-ball of $\mathbf{x}^0$ (Madry et al., 2017):

$$\mathbf{x}^k = \prod \left( \mathbf{x}^{k-1} + \alpha \cdot \text{sign}(\nabla_{\mathbf{x}} \ell(h_{\boldsymbol{\theta}}(\mathbf{x}^{k-1}), y)) \right) \tag{2}$$

where $\alpha$ is the step size, $\prod(\cdot)$ is the orthogonal projection function onto $\{\mathbf{x}' : \|\mathbf{x}^0 - \mathbf{x}'\| \le \epsilon\}$, and $\mathbf{x}^k$ is the adversarial example at $k$-th step.

However, a major problem prohibiting adversarial training to be practically applicable is the huge computational burden associated with the inner maximization steps: we need to iteratively solve the inner maximization problem to find good adversarial examples for DNN to be robust. Recently, to accelerate adversarial training, a few methods have been proposed. For example, YOPO estimated the gradient on the input by only propagating the first layer (Zhang et al., 2019a), and parallel adversarial training utilized multiple graphics computation units (GPUs) for acceleration (Bhat & Tsipras, 2019; Shafahi et al., 2019). A common drawback of these methods are their implementation complexity or their need for multiple GPUs for acceleration. On the other hand, an empirical observation made by (Wang et al., 2019), indicated that we might not need to find good solutions to the inner maximization at the initial stages of adversarial training to achieve even better robustness.

In line of such observations, in this paper we propose a simple yet effective Annealing Mechanism for Adversarial Training Acceleration, which we call **Amata**, that gradually controls the degree of which the inner maximization is solved as the training proceeds. This annealing algorithm only takes 1/3 to 1/2 the time to achieve comparable or even better robustness with the addition of only two lines of code. To motivate the present approach, we develop a general formulation of adversarial training as an optimal control problem, from which an approximate optimality criterion can be derived based on the Pontryagin's maximum principle. This criterion is related to but different from the empirical FOSC criterion (Wang et al., 2019) that improves the robustness of adversarial training by determining the number of adversarial training steps according to the empirical criterion. We will also demonstrate in our experiments that unlike FOSC, algorithms derived from our proposed criterion leads to acceleration.

Our contributions are as follows:

1. We propose an adversarial training algorithm, Amata, that gradually increases the number of iterations and decreases the step size for the inner maximization. We show that comparable or even better robustness can be achieved with much reduced computational overhead. We also prove the convergence of our algorithm.

2. To obtain more theoretical insight, we develop a general formulation of adversarial training subject to hyper-parameters in the inner maximization loop as an optimal control problem. This includes the PGD-based methods considered in this work, but potentially encompasses a larger class of annealed or adaptive adversarial training algorithms.

3. Using the Pontryagin's maximum principle from optimal control theory, we provide a principled criterion to measure the near-term optimality of an annealing schedule for inner maximization, thereby guiding and validating our proposed algorithm in balancing the trade-off between robustness and computational cost.

## 2 ACCELERATING ADVERSARIAL TRAINING BY ANNEALING MECHANISM

In this section, we will first introduce the proposed adversarial training algorithm, which aims to balance the computational cost and the accuracy of solving inner maximization. A proof of the convergence of the algorithm can be found in the Appendix. We then develop a general optimal control formulation of adversarial training and derive a criterion based on the Pontryagin's maximum principle (Boltyanskii et al., 1960). The criterion is shown to be effective in quantifying adversarial training performance both on accuracy and computational efficiency, justifying the superiority of the proposed algorithm.

### 2.1 PROPOSED ANNEALING ADVERSARIAL TRAINING ALGORITHM

The proposed adversarial training algorithm Amata is found in Algorithm 1. Compared with PGD, we only need to add two lines of code that are shown in blue. The intuition behind Amata is that, at

the initial stage, the neural network focus on learning features, which might not require very accurate adversarial examples. Therefore, we only need coarse approximations of the inner maximization problem solutions. With this consideration, we set a small number of update steps but with a large step size for inner maximization, and then gradually increase $K$ and decreases $\alpha$ to improve the quality of inner maximization solutions. This adaptive annealing mechanism would largely reduce the computational cost in the early iterations while still maintaining reasonable accuracy for the entire optimization.

---

**Algorithm 1** Amata: an annealing mechanism for adversarial training acceleration

---

**Input:** $T$:training epochs; $K_{\min}$: the minimum number of adversarial perturbations; $K_{\max}$: the maximum number of adversarial perturbations; $\boldsymbol{\theta}$: parameter of neural network to be adversarially trained; $\mathcal{B}$:mini-batch; $\alpha$: adversarial training time step; $\eta$: learning rate of neural network parameters. $\tau$: constant, maximum perturbation:$\epsilon$.
**Initialization** $\boldsymbol{\theta} = \boldsymbol{\theta}^0$
**for** $t = 0$ to $T - 1$ **do**
$\quad$ Compute the annealing number of adversarial perturbations:

$$K^t = K_{\min} + (K_{\max} - K_{\min}) \cdot \frac{t}{T}$$

$\quad$ Compute adversarial perturbation step size: $\alpha^t = \frac{\tau}{K^t}$
$\quad$ **for** each mini-batch $\mathbf{x}_{\mathcal{B}}^0$ **do**
$\quad\quad$ **for** $k = 1$ to $K^t$ **do**
$\quad\quad\quad$ Compute adversarial perturbations:

$$\mathbf{x}_{\mathcal{B}}^k = \mathbf{x}_{\mathcal{B}}^{k-1} + \alpha_t \cdot \mathrm{sign}(\nabla_{\mathbf{x}} \ell(h_{\boldsymbol{\theta}}(\mathbf{x}_{\mathcal{B}}^k), y)$$
$$\mathbf{x}_{\mathcal{B}}^k = \mathrm{clip}(\mathbf{x}_{\mathcal{B}}^k, \mathbf{x}_{\mathcal{B}}^0 - \epsilon, \mathbf{x}_{\mathcal{B}}^0 + \epsilon)$$

$\quad\quad$ **end for**
$\quad\quad$ $\boldsymbol{\theta}_{t+1} = \boldsymbol{\theta}_t - \eta \nabla_{\boldsymbol{\theta}} \ell(h_{\boldsymbol{\theta}_t}(\mathbf{x}_{\mathcal{B}}^{K^t}), y)$
$\quad$ **end for**
**end for**
Collect $\boldsymbol{\theta}_T$ as the parameter of adversarially-trained neural network.

---

In the following section, we develop a general formulation of adversarial training based on optimal control theory and derive a novel criterion to quantify the optimality of a training strategy, taking into account the trade-off between accuracy and efficiency. Importantly, we show that our proposal, Amata, performs favorably under this criterion, thus providing theoretical justification for our approach.

## 2.2 Optimal Control Formulation of Adversarial Training

In essence, the PGD-based adversarial training algorithm (Madry et al., 2017) represents a sequence of relaxations of the original min-max problem (1). First, the outer minimization over $\boldsymbol{\theta}$ is replaced by gradient descent. Then, a full solution of the inner maximization is replaced by a number of PGD steps in the steepest ascent direction. Consequently, a natural question is how to choose the number of steps, the step size or any other hyper-parameters associated with this relaxed version of the inner maximization, and how their choices affect the performance of the overall algorithm. This is the central point of analysis in this paper and the basis of the algorithm proposed. It turns out that this question can be systematically formulated in the framework of optimal control theory (Bertsekas, 1995). We now introduce the general setup of our problem and establish connections with optimal control, and in particular the classical maximum principle in the calculus of variations.

For simplicity of presentation, let us consider just one fixed input-label pair $(\mathbf{x}^0, y)$, since the $n$-sample case is similar. The original min-max adversarial training problem is

$$\min_{\boldsymbol{\theta}} \max_{\{\mathbf{z}: \|\mathbf{z} - \mathbf{x}^0\| \leq \epsilon\}} \ell(h_{\boldsymbol{\theta}}(\mathbf{z}), y). \tag{3}$$

The first relaxation is to replace the outer minimization with gradient descent so that we obtain the iteration

$$\boldsymbol{\theta}_{t+1} = \boldsymbol{\theta}_t - \eta \nabla_{\boldsymbol{\theta}} \max_{\{\mathbf{z}: \|\mathbf{z}-\mathbf{x}^0\| \leq \epsilon\}} \ell(h_{\boldsymbol{\theta}_t}(\mathbf{z}), y). \tag{4}$$

Then, the remaining maximization in each outer iteration step is replaced by an abstract algorithm

$$\mathcal{A}_{\mathbf{u},\boldsymbol{\theta}} : \mathcal{R}^d \to \mathcal{R}^d \tag{5}$$

which solves the inner maximization approximately. Here, we assume that the algorithm depends on the parameters of our neural network, as well as hyper-parameters $\mathbf{u}$ which takes values in a closed subset $G$ of a Euclidean space. No further assumptions is placed on $G$, which may be a continuum, a discrete set, or even a finite set.

This relaxation leads to the following general iterations[1]

$$\boldsymbol{\theta}_{t+1} = \boldsymbol{\theta}_t - \eta \nabla_{\boldsymbol{\theta}} \ell(h_{\boldsymbol{\theta}_t}(\mathcal{A}_{\boldsymbol{\theta}_t, \mathbf{u}_t}), y). \tag{6}$$

Observe that Algorithm 1 is a particular realization of (6) where $\mathcal{A}_{\theta_t, u_t}$ represents the inner PGD loop and $\mathbf{u}_t = \{\alpha_t, K^t\}$ are the hyper-parameters we pick at each $t$. Nevertheless, Equation (6) represents a general formulation of an inner-outer loop adversarial training algorithm. For small $\eta$, we can replace (6) by an ordinary differential equation with the identification $s \approx t\eta$:

$$\dot{\boldsymbol{\theta}}_s = -\nabla_{\boldsymbol{\theta}} \ell(h_{\boldsymbol{\theta}_s}(\mathcal{A}_{\boldsymbol{\theta}_s, \mathbf{u}_s}), y). \tag{7}$$

Formulated this way, it is immediately clear that we can approach this problem from a control perspective: the dynamics of the trainable parameters are given by (7) and the hyper-parameter choices at each inner algorithm $\mathbf{u}_s$ are the controls.

Our goal is two-fold: on a training interval $[T_1, T_2]$ in the outer loop, we want to minimize some loss under adversarial training measured by a real-valued function $\Phi(\boldsymbol{\theta})$ while minimizing training cost associated with each inner algorithm loop under the hyper-parameter $\mathbf{u}$, which is measured by another real-valued function $R(\mathbf{u})$. We thus arrive at the general optimal control formulation of our problem:

$$\min_{\mathbf{u}_{T_1:T_2} \in L^\infty_{T_1:T_2}} \Phi(\boldsymbol{\theta}_T) + \int_{T_1}^{T_2} R(\mathbf{u}_s) ds \qquad \text{subject to:} \tag{8}$$
$$\dot{\boldsymbol{\theta}}_t = F(\boldsymbol{\theta}_s, \mathbf{u}_s) \qquad \text{where} \qquad F(\boldsymbol{\theta}_s, \mathbf{u}_s) := -\nabla_{\boldsymbol{\theta}} \ell(h_{\boldsymbol{\theta}_s}(\mathcal{A}_{\boldsymbol{\theta}_s, \mathbf{u}_s}), y),$$

where we have defined the shorthand $\mathbf{u}_{T_1:T_2} = \{\mathbf{u}_s : s \in [T_1, T_2]\}$ and $L^\infty_{T_1:T_2} := L^\infty([T_1, T_2], G)$. In this paper, we take $\Phi$ to be the DNN's prediction loss given the adversarial example (adversarial robustness), and $R$ is set as $\gamma K^t$ where $\gamma$ is the coefficient for adversarial robustness and training time trade-off. This is to account for the fact that when $K^t$ increases, the cost of the inner loop training increases accordingly. The integral over $t$ of $R$ is taken so as to account for the total computational cost corresponding to a choice of hyper-parameters $\{\mathbf{u}_s\}$. The objective function taken as a sum serves to balance the adversarial robustness and computational cost, with $\gamma$ a balancing coefficient.

### 2.3 PONTRYAGIN'S MAXIMUM PRINCIPLE

In the last section, the problem of choosing hyper-parameters in the inner loops of adversarial training has been formulated as an optimal control problem in (8). Now, we show how this connection can help us design and validate algorithms.

A classical result in calculus of variations gives the following necessary conditions for optimality. For more background on the theory of calculus of variations and optimal control, we refer the reader to Boltyanskii et al. (1960); Bertsekas et al. (1995).

---

[1]Here we assume that the gradient with respect to $\theta$ is the partial derivative with respect to the parameters of the network $h_\theta$ and $\theta_t$ in $\mathcal{A}_{\theta_t, \mathbf{u}_t}$ is held constant. This is the case for the PGD algorithm. Alternatively, we can also take the total derivative, but this leads to different algorithms.

**Theorem 1** (Pontryagin's Maximum Principle (PMP) (Boltyanskii et al., 1960)). *Let* $\mathbf{u}^*_{T_1:T_2} \in L^\infty_{T_1,T_2}$ *be a solution to* (8). *Suppose* $F(\boldsymbol{\theta}, \mathbf{u})$ *is Lipschitz in* $\boldsymbol{\theta}$ *and measurable in* $\mathbf{u}$. *Define the Hamiltonian function*

$$H(\boldsymbol{\theta}, \mathbf{p}, \mathbf{u}) = \mathbf{p}^T F(\boldsymbol{\theta}, \mathbf{u}) - R(\mathbf{u}) \tag{9}$$

*Then, there exists an absolutely continuous co-state process* $\mathbf{p}^*_{T_1:T_2}$ *such that*

$$\dot{\boldsymbol{\theta}}^*_s = F(\boldsymbol{\theta}^*_s, \mathbf{u}^*_s) \qquad\qquad \boldsymbol{\theta}^*_{T_1} = \boldsymbol{\theta}_{T_1} \tag{10}$$

$$\dot{\mathbf{p}}^*_s = -\nabla_{\boldsymbol{\theta}} H(\boldsymbol{\theta}^*_s, \mathbf{p}^*_s, \mathbf{u}^*_s) \qquad\qquad \mathbf{p}^*_{T_2} = -\nabla_{\boldsymbol{\theta}} \Phi(\boldsymbol{\theta}^*_{T_2}) \tag{11}$$

$$H(\boldsymbol{\theta}^*_s, \mathbf{p}^*_s, \mathbf{u}^*_s) \geq H(\boldsymbol{\theta}^*_s, \mathbf{p}^*_s, \mathbf{v}) \qquad\qquad \text{for all } \mathbf{v} \in G, s \in [T_1, T_2] \tag{12}$$

In short, the maximum principle says that a set of optimal parameter choices $\{\mathbf{u}^*_s\}$ (in our specific application, these are the optimal choices of inner-loop hyper-parameters as the training proceeds) must globally maximize the Hamiltonian defined above for *each* outer iteration. This statement is especially appealing for our application because unlike first-order gradient conditions, the PMP holds even when our hyper-parameters can only take a discrete set of values, or when there are non-trivial constraints amongst them. The maximization criterion holds generally under these conditions. Moreover, we now show that it gives us a quantitative measure of deviation from optimality, from which we can analyze and design algorithms.

**Quantitative Measure of Sub-optimality.** Given any hyper-parameter choice $\mathbf{u}_{T_1:T_2}$ over the training interval, let us define its "distance" from optimality as

$$C(\mathbf{u}_{T_1:T_2}) := \frac{1}{T_2 - T_1} \int_{T_1}^{T_2} \max_{\mathbf{v} \in G} H(\boldsymbol{\theta}^{\mathbf{u}}_s, \mathbf{p}^{\mathbf{u}}_s, \mathbf{v}) - H(\boldsymbol{\theta}^{\mathbf{u}}_s, \mathbf{p}^{\mathbf{u}}_s, \mathbf{u}_s) ds \tag{13}$$

where $\{\boldsymbol{\theta}^{\mathbf{u}}_s, \mathbf{p}^{\mathbf{u}}_s : s \in [T_1, T_2]\}$ represents the solution of the equations (10) and (11) with $\mathbf{u}_s$ in place of $\mathbf{u}^*_s$. Observe that $C(\mathbf{u}_{T_1:T_2}) \geq 0$ for any $\mathbf{u}_{T_1:T_2}$ with equality if and only if $\mathbf{u}_{T_1:T_2}$ satisfies the PMP for almost every $s \in [T_1, T_2]$. Hence, $C$ can be used as a measure of sub-optimality. When $C$ is small, our annealing strategy $\{\mathbf{u}_s\}$ is close to at least an locally optimal strategy, where as when it is large, our annealing strategy is far from an optimal one.

**One-step Approximation and Adversarial Training Criterion.** Equation (13) requires information on the entire training interval and may be expensive to compute. In this paper, we use a one-step approximation where we take $T_1 = t$ (current iteration) and $T_2 = t + \eta$ with $\eta \ll 1$. In this small interval, we can also take $\mathbf{u}_s$ to be constant and thus equal to some $\mathbf{u}$. This is in some sense a greedy approximation, where we evaluate in the immediate short term the optimality of a piece-wise constant choice of hyper-parameters. From (13) we then obtain via a Taylor expansion and our particular choices of $\Phi$ and $R$ (See Appendix D)

$$C(\mathbf{u}_{t:t+\eta}) = C(\mathbf{u}_t, t) + o(1) \text{ with}$$

$$C(\mathbf{u}_t, t) \equiv C(\alpha_t, K^t, t) \approx \max_{\alpha, K} \left\{ \|\nabla_{\boldsymbol{\theta}} \ell(h_{\boldsymbol{\theta}_t}[\mathcal{A}_{\boldsymbol{\theta}_t, \alpha, K}(x)], y)\|^2 - \gamma K \right\}$$
$$- \left( \|\nabla_{\boldsymbol{\theta}} \ell(h_{\boldsymbol{\theta}_t}[\mathcal{A}_{\boldsymbol{\theta}_t, \alpha_t, K^t}(x)], y)\|^2 - \gamma K^t \right), \tag{14}$$

with $\mathcal{A}_{\boldsymbol{\theta}_t, \alpha, K}$ denoting the inner PGD loop starting form $x$ with $K$ steps and step size $\alpha$. Criterion (14) is a greedy version of the general criterion derived from the maximum principle. It can be used to either evaluate the near-term sub-optimality of some choice of hyper-parameters $\mathbf{u}$, or to find an approximately optimal hyperparameter greedily by solving $C(\mathbf{u}, t) = 0$ for $\mathbf{u}$, which amounts to maximizing the first term. In this paper, we use Bayesian optimization[2] to perform the maximization in (14) to evaluate and select strategies from the controllable space $G$.

**Remark.** By applying the method of successive approximations (MSA) (Chernousko & Lyubushin, 1982; Li et al., 2017; Li & Hao, 2018; E et al., 2019), we can also obtain a solution of the PMP by iterating

$$\mathbf{u}^{k+1}_s = \arg\max_{\mathbf{v} \in G} H(\boldsymbol{\theta}^{\mathbf{u}^k}_s, p^{\mathbf{u}^k}_s, \mathbf{v}), \qquad s \in [T_1, T_2]. \tag{15}$$

---

[2] Implementations can be found in https://github.com/hyperopt/hyperopt

Subsequent work will explore adaptive adversarial training acceleration based on this approach. We remark that although we appeal to control theory tools here, the application scenario is very different from prior work in Li et al. (2017); Li & Hao (2018); E et al. (2019). For the latter, the controlled dynamics are idealized forward propagation through a deep neural network and the controls are the trainable weights of the network. In contrast, in the current setting the controlled dynamics are the evolution of weights under gradient descent in the outer loop and controls are the hyper-parameters for the adversarial perturbation inner loop.

**Comparison with FOSC criterion:** Wang et al. (2019) proposed an empirical criterion to measure the convergence of inner maximization:

$$\text{FOSC}(\mathbf{x}) = \epsilon \left\| \nabla_{\mathbf{x}} \ell(h_{\boldsymbol{\theta}}(\mathbf{x}), y) \right\| - \langle \mathbf{x} - \mathbf{x}^0, \nabla_{\mathbf{x}} \ell(h_{\boldsymbol{\theta}}(\mathbf{x}), y) \rangle \tag{16}$$

There are some similarities between our criterion and FOSC when we do not consider the computational cost term $R$. For example, when the stationary saddle point is achieved, both our criterion and FOSC reach the minimum. However, our proposed criterion is quite different from FOSC in the following aspects:

1. Our criterion is derived from the optimal control theory, whereas FOSC is concluded from empirical observations.

2. Our criterion takes computation costs into consideration, whereas FOSC only considers the convergence of adversarial training.

3. Our criterion is based on the gradient of DNN parameters, whereas FOSC is based on the gradient of the input. Measuring the gradient of DNN parameters is arguably more suitable for considering robustness-training time trade-off as the variance of the DNN parameters is much larger than the input during training.

## 2.4 NUMERICAL JUSTIFICATION OF AMATA

For simple linear-quadratic control problems, we can typically derive some analytical representations of an optimal control. However, for highly-nonconvex DNNs, we can only use a numerical form of the optimal control criterion (Equation 14) to analyze Amata for robustness and computational efficiency trade-off. We use the LeNet neural network [3] for MNIST classification as an example.

As a sanity check, we first consider only adversarial robustness and neglect computational cost by setting $\gamma = 0$ and use the criterion to evaluate different adversarial training settings for PGD. Recalling Algorithm 1, we will vary the other hyper-parameters in the inner loop: $\alpha$ (step-size of PGD) and $K$ (number of PGD steps). In each case, we compute the optimal control criterion values on the training dataset and evaluate the adversarially trained networks' performance against PGD-40 attack on the test dataset.

**Sanity check 1** In this case, we fix $K = 5$ and set $\alpha$ in $[0.01, 0.02, 0.03, 0.04, 0.05]$.

**Sanity check 2** In this case, we fix $\alpha = 0.01$ and set $K$ in $[5, 10, 20, 30, 40, 50]$.

The results are shown in Figure 1 (a)(b), where robustness is the prediction accuracy of the trained networks under PGD-40 attack on the test dataset.

From these results, we can observe that the robustness and the criterion value $C$ are negatively correlated under varying $\alpha$ and $K$, as expected.

Now, we will use the same setting as sanity checks but with different values of $\gamma$ to numerically analyze the trade-off between robustness and training time for Amata and PGD.

**Setting 1: fixed $\alpha$, varying $K$ and $\gamma$:** in this case, we set $\alpha$ as 0.01, $K$ in $[5, 20, 40]$ and $\gamma$ in $[0.02, 0.04, 0.06, 0.08]$. The result for Setting 1 is shown in Figure 1 (c). From this result, we can observe that with the increase of $\gamma$, reducing the computation time becomes more important. For example, when $\gamma$ is 0.02, using PGD-40 is the best strategy (the lowest $C$ value). When $\gamma$ increases to 0.06 or 0.08, using PGD-20 achieves the best trade-off under the criterion.

---

[3]Implementations can be found in https://github.com/pytorch/examples/blob/master/mnist/main.py

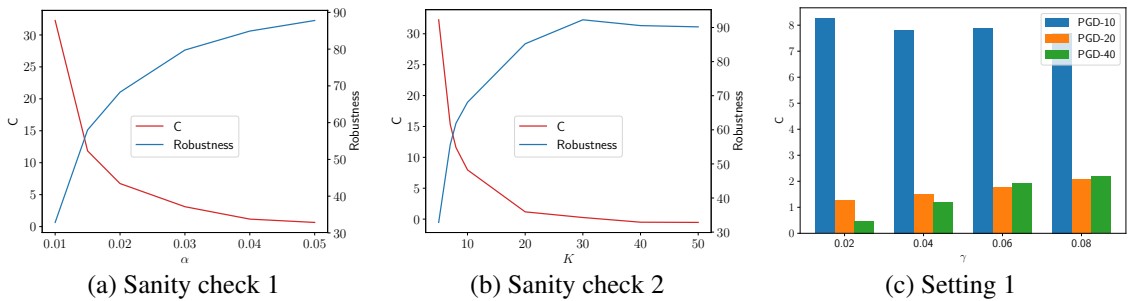

|  | |  |
|:---:|:---:|:---:|
| (a) Sanity check 1 | (b) Sanity check 2 | (c) Setting 1 |

**Figure 1:** Sanity checks of the optimal control criterion ($\gamma = 0$).

**Table 1:** Comparison of adversarial training strategies.

| Strategy | C($\gamma = 0.04$) | C($\gamma = 0.08$) | Robustness | Training time (seconds) |
|---|---|---|---|---|
| Amata($K_{min} = 5, K_{max} = 40$) | **0.54** | **1.38** | **91.47%** | 697.73 |
| Amata($K_{min} = 10, K_{max} = 40$) | **0.68** | **1.53** | **91.46%** | 760.16 |
| PGD-10 | 7.82 | 7.70 | 68.07% | 307.57 |
| PGD-20 | 1.52 | 2.09 | 85.23% | 567.11 |
| PGD-40 | 1.20 | 2.19 | 90.56% | 1086.31 |

Next, we set $\gamma$ as 0.04 and 0.08 and use the criterion to evaluate the Amata and compare it with other settings of PGD. It can be seen that Amata is better than PGD with fixed numbers of perturbation steps. Note that Amata also achieves better robustness than the highest robustness of PGD settings (PGD-40) with $91.47\%$ adversarial accuracy compared to $90.56\%$ adversarial accuracy.

**Remark:** Although computing the exact optimal control strategy for DNN adversarial training is inapplicable for real-time applications, with the criterion derived from the PMP, we are able to numerically compare the optimality of different adversarial training strategies. From this numerical evaluation, we have demonstrated that the proposed Amata algorithm is close to an optimal adversarial training strategy, or at least one that satisfies the maximum principle. We will show that our algorithm can achieve similar or even better adversarial accuracy much faster with empirical experiments on popular DNN models later in Section 3.

## 3 EXPERIMENTS

To demonstrate the effectiveness of Amata, we conduct experiments on MNIST and CIFAR10. We find that the models trained with Amata have comparable or even better performance with that of the PGD adversarial training, but with much less computational cost. For our experiment, we use PyTorch 1.0.0 and a GTX1080 Ti GPU. We evaluate the adversarial trained networks against PGD and Carlini-Wagner (CW) attack (Carlini & Wagner, 2017).

### 3.1 MNIST CLASSIFICATION

We consider the standard MNIST classification using the "smallCNN" architecture that was used in (Zhang et al., 2019b). We set $\tau$ to be 0.4 for Amata. In the experiment, We achieve 96% adversarial accuracy within 275 seconds while it takes PGD-40 1019 seconds to reach the same level. We also experimented with the FOSC adversarial training method in (Wang et al., 2019) and found that it took FOSC 3663 seconds to achieve the same level, which confirms our previous analysis that our optimal control criterion is more suitable for accelerating adversarial training. The error-time curve is shown in Figure 2 (a). Then, we run all methods for 57 epochs for full convergence. The robustness of convergence results are shown in Table 2. From Table 2, we can see that naively reducing the number of adversarial perturbations will hurt the robustness of DNNs.

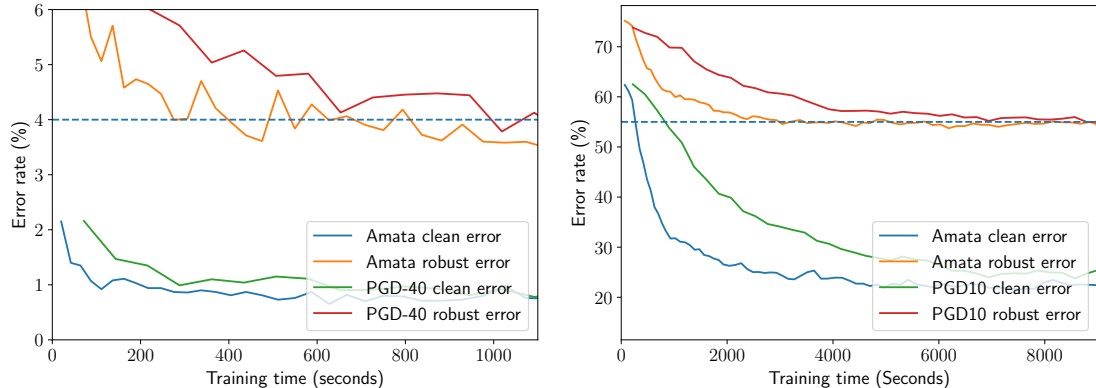

**Figure 2: Left:** MNIST result. Training time against PGD-40 attack. We use Amata with the setting $K_{min} = 10$ and $K_{max} = 40$. **Right:** CIFAR10 result. Training time against PGD-20 attack. We use Amata with the setting $K_{min} = 2$ and $K_{max} = 10$.

**Table 2:** MNIST adversarial training results.

| Training methods | Clean accuracy | PGD-40 Attack | CW Attack | Time (Seconds) |
|---|---|---|---|---|
| ERM | 99.52% | 5.58% | 0.10% | 268.10 |
| PGD-5 | **99.58%** | 80.99% | 0.31% | 709.21 |
| PGD-10 | 99.57% | 93.8% | 1.06% | 1234.30 |
| PGD-40 | 99.51% | 97.02% | 93.49% | 4331.81 |
| FOSC(Wang et al., 2019) | 99.51% | 97.02% | 81.79% | 14928.53 |
| Amata($K_{min} = 5, K_{max} = 40$) | 99.53% | **97.11%** | 86.92% | 2533.28 |
| Amata($K_{min} = 10, K_{max} = 40$) | 99.45% | 96.97% | **94.37%** | 2675.64 |

## 3.2 CIFAR10 CLASSIFICATION

For the more complex CIFAR10 classification task, we use the PreAct-Res-18 network (Madry et al., 2017). We set $\tau$ to be 20/255 for Amata. The proposed Amata method took 3045 seconds to achieve less than 55% robust error while for PGD-10, it took 6944 seconds. We also tested FOSC adversarial training and found that it took 8385 seconds to reach the same level, which is a bit longer than PGD-10. The error-time curve is shown in Figure 2 (b). We further run all methods for 100 epochs for convergence. The results are shown in Table 3.

**Table 3:** CIFAR10 PreAct-Res-18 adversarial training results.

| Training methods | Clean accuracy | PGD-20 Attack | CW Attack | Time (Seconds) |
|---|---|---|---|---|
| ERM | 94.75% | 0.0% | 0.23% | 2099.58 |
| PGD-2 | 90.16% | 31.70% | 13.36% | 6913.36 |
| PGD-10 | 85.27% | 47.31% | 51.73% | 23108.10 |
| FOSC(Wang et al., 2019) | 85.47% | **48.04%** | **53.65%** | 26126.98 |
| Amata($K_{min} = 2, K_{max} = 10$) | 85.52% | 47.62% | 52.94% | 14308.96 |

## 4 CONCLUSION

We have proposed a modification amounting to two lines of code to PGD adversarial training that achieves comparable or even better robustness with only 1/3 to 1/2 the computational cost over a variety of benchmarks. Moreover, a general optimal control formulation of adversarial training with hyper-parameters is developed to analyze this procedure and justify its superior performance through a numerical criterion based on the Pontryagin's maximum principle. A proof of convergence of our algorithm is also provided in Appendix C. As a point of future work, we will explore adaptive methods for adversarial training based on the optimal control formulation we introduced here. This approach can also lead to algorithms that can be combined with YOPO or parallel adversarial training methods for maximal efficiency. An example is shown in Appendix B.3.

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

# A  APPENDIX: ADDITIONAL EXPERIMENT DETAILS

We use similar experiment settings as in (Zhang et al., 2019a).

## A.1  MNIST CLASSIFICATION

In this experiment, for PGD adversarial training, we set the adversarial constraint $\epsilon$ as 0.3, the step size as 0.01. For CW attack results in this experiment, we set the parameter eps to be 100 and run for 100 iterations in this reference implementation https://github.com/xuanqing94/BayesianDefense/blob/master/attacker/cw.py. For display the error-time curve, we use the smoothing function used in Tensorboard for all methods. The smoothing parameter is set as 0.09. For outer minimization, we use the stochastic gradient descent method and set the learning rate as 0.1, the momentum as 0.9, and the weight decay as 5e-4.

## A.2  CIFAR10 CLASSIFICATION

In this experiment, for PGD adversarial training, we set the adversarial constraint $\epsilon$ as 8/255, the step size as 2/255. For CW attack results in this experiment, we set the parameter eps to be 0.5 and run for 100 iterations in this reference implementation https://github.com/xuanqing94/BayesianDefense/blob/master/attacker/cw.py. For display the error-time curve, we use the smoothing function used in Tensorboard for all methods. The smoothing parameter is set as 0.6. For outer minimization, we use the stochastic gradient descent method and set the learning rate as 5e-2, the momentum as 0.9, and the weight decay as 5e-4. We also use a piece-wise constant learning rate scheduler in PyTorch by setting the milestones at the 75-th and 90-th epoch with a factor of 0.1.

# B    APPENDIX:ADDITIONAL EXPERIMENT RESULTS

## B.1    WIDERESNET-34 RESULTS

We experiments using WideResNet-34 networks for CIFAR10 classification. We use the same experiment setting as the CIFAR10 classification task in the paper. We run all methods for 100 epochs for full convergence. The result is shown in Table 4.

**Table 4:** CIFAR10 WideResNet-34 adversarial training results.

| Training methods | Clean accuracy | PGD-20 Attack | CW Attack | Time (Seconds) |
|---|---|---|---|---|
| PGD-10 | 85.80% | 45.38% | 30.82% | 150653.40 |
| Amata($K_{min} = 2, K_{max} = 10$) | 85.77% | 30.42% | **52.94%** | 90945.15 |

## B.2    ADDITIONAL MNIST RESULTS

We further plot the MNIST results in Figure 2 but with the x-axis denoting the number of training epochs. The results are shown in Figure 3. From this figure, we can see that the time-saving of our method is mainly due to the savings in the adversarial examples generation step. This is expected as if we achieve faster convergence rates with regard to the number of epochs, it may indicate that the adversarial examples generated by the proposed mechanism is significantly weaker and the outer minimization step converges faster.

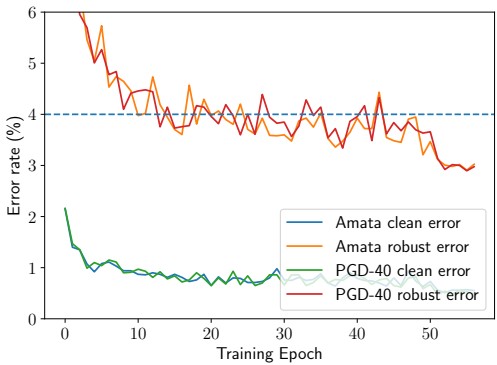

**Figure 3:** MNIST result. Training epoch against PGD-40 attack. We use Amata with the setting $K_{min} = 10$ and $K_{max} = 40$.

## B.3    AMATA+YOPO

To demonstrate that our method can be incorporated in other acceleration method to further improve the performance, we implement the Amata in YOPO MNIST classification experiment[4]. For Amata incorporation, we gradually increase the $K$ and decrease $\sigma$ in the codes that is similar to the case of modifying the PGD algorithm. We run Amata+YOPO and YOPO for 40 epochs as in the original setting. The results are shown in Figure 4. From Figure 4, we can see that Amata+YOPO takes 294 seconds to reach the adversarial accuracy of 94%, which is only around half of the time for YOPO to achieve the same adversarial accuracy. It is also worth noting that Amata+YOPO achieves even better adversarial accuracy when converged. From this example, we can see that Amata can be seamlessly incorporated in other adversarial training acceleration algorithm to provide further acceleration.

---

[4]https://github.com/a1600012888/YOPO-You-Only-Propagate-Once/tree/82c5b902508224c642c8d0173e61435795c0ac42/experiments/M 5-10

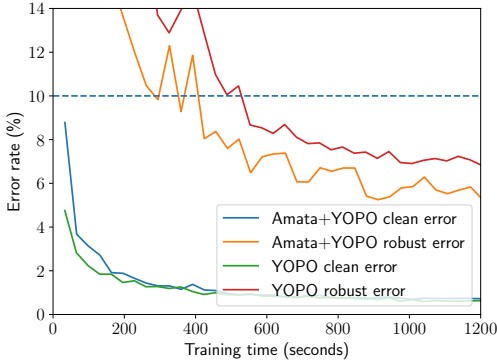

**Figure 4:** MNIST result. Training epoch against PGD-40 attack. We use Amata with the setting $K_{min} = 2$ and $K_{max} = 5$.

## B.4 EVALUATION AGAINST DIFFERENT SETTINGS OF PGD

The reason we choose PGD-40 for evaluation is that is a strong adversarial attack that is also used in other papers (Zhang et al., 2019a). We also try other settings, such as PGD-20 and PGD-60. Other settings are the same as our original settings in MNIST classification experiment. The results are shown in Figure 5. We can see that compared with the case of PGD-40 for evaluation, both Amata and PGD-40 are able to achieve above the 96% adversarial accuracy but at different speeds. In the case of PGD-20, Amata takes around 28% training time of the PGD-40 to achieve the 96% adversarial accuracy. In the case of PGD-60, Amata takes around 60% training time of the PGD-40 to achieve the 96% adversarial accuracy.

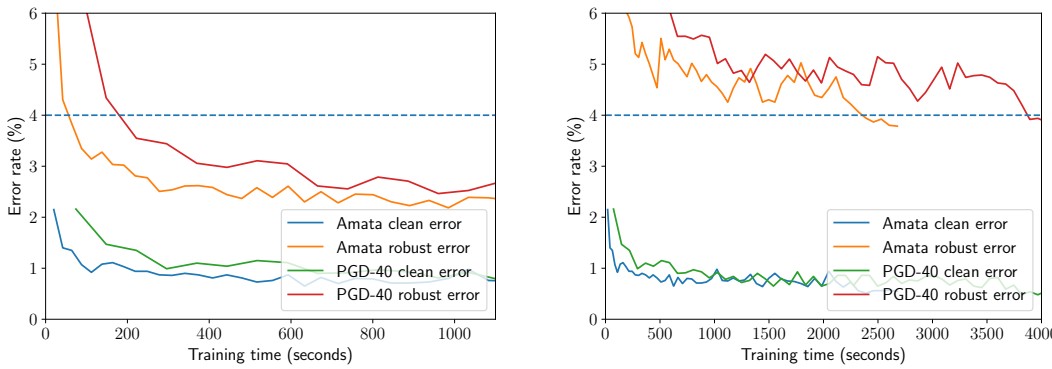

**Figure 5: Left:** MNIST result. Training epoch against PGD-20 attack.**Right:** MNIST result. Training epoch against PGD-60 attack.

## B.5 OTHER DECAY SCHEME FOR CONTROLLING ADVERSARIAL TRAINING

Finding better decay scheme is an interesting future direction. However, using more complex schemes may introduce extra parameters, while one of the major challenges for machine learning methods to be practically useful for industry practitioners is the complexity of the method. We tried another exponential scheme:

$$K_t = K_{min} + (K_{max} - K_{min}) \cdot (1 - \exp^{-\eta \cdot t})/(1 - \exp^{-\eta \cdot T}) \tag{17}$$

where $K_t$ is the number of PGD adversarial training steps at $t$-th epoch, $K_{min}$ is the minimum number of PGD steps in the beginning, $K_{max}$ is the maximum number of PGD steps in the last epoch, $\eta$ is the hyper-parameter controlling the shape of the scheme. This design ensures that when

$K_0 = K_{min}$ and $K_T = K_{max}$ to make the exponential decay scheme comparable with the linear scheme used in our paper. However, we tried different parameters of eta ranging from 0.1 to 5, but did not performance improvements. The results are shown in Figure 6. The "Amata (Exp eta)" in the legend denotes the exponential decay scheme and the "Amata" in the legend denotes the original linear decay scheme. From Figure 6, we can see that the the new exponential decay scheme cannot outperform the linear decay scheme.

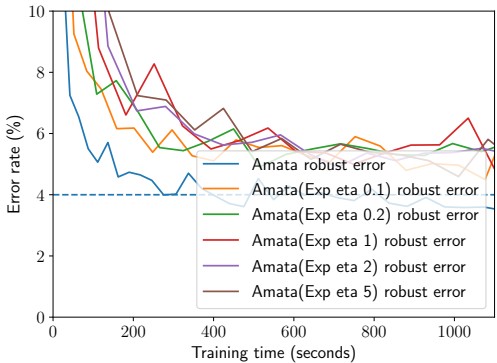

**Figure 6:** MNIST result. Training epoch against PGD-40 attack. We use Amata with the setting $K_{min} = 2$ and $K_{max} = 5$.

## C  APPENDIX:PROOF OF CONVERGENCE

We provide a convergence analysis of our proposed Amata algorithm for solving the min-max problem. The proof of convergence in this paper largely follows (Wang et al., 2019; Sinha et al., 2018). First, we will introduce some notations first for clarity. We denote $\mathbf{x}_i^*(\boldsymbol{\theta}) = \arg\max_{\mathbf{x}_i \in \mathcal{X}^i} f(\boldsymbol{\theta}, \mathbf{x}_i)$ where $f(\boldsymbol{\theta}, \mathbf{x}_i)$ is a short hand notation for the classification loss function, $\mathcal{X}^i = \{\mathbf{x} | \|\mathbf{x} - \mathbf{x}_i^0\| \leq \epsilon\}$, and $\bar{f}_i(\boldsymbol{\theta}) = \max_{\mathbf{x}_i \in \mathcal{X}^i} f(\boldsymbol{\theta}, \mathbf{x}_i)$, then $\tilde{\mathbf{x}}_i(\boldsymbol{\theta})$ is a $\delta$-approximate solution by our algorithm to $\mathbf{x}_i^*(\boldsymbol{\theta})$, if it satisfies that:

$$\|\tilde{\mathbf{x}}_i(\boldsymbol{\theta}) - \mathbf{x}_i^*(\boldsymbol{\theta})\| \leq \delta \tag{18}$$

In addition, denote the objective function in Equation 1 by $L_S(\boldsymbol{\theta})$ , and its gradient by $\nabla L_S(\boldsymbol{\theta}) = \frac{1}{n}\sum_{i=1}^n \nabla \bar{f}_i(\boldsymbol{\theta})$. Let $g(\boldsymbol{\theta}) = \frac{1}{|\mathcal{B}|}\sum_{i\in\mathcal{B}} \nabla_{\boldsymbol{\theta}}\bar{f}(\boldsymbol{\theta})$ be the stochastic gradient of $L_S(\boldsymbol{\theta})$, where $\mathcal{B}$ is the mini-batch. Then, we have $\mathbb{E}[g(\boldsymbol{\theta})] = \nabla L_S(\boldsymbol{\theta})$. Let $\nabla_{\boldsymbol{\theta}}f(\boldsymbol{\theta}, \tilde{\mathbf{x}}(\boldsymbol{\theta}))$ be the gradient of $f(\boldsymbol{\theta}, \tilde{\mathbf{x}}(\boldsymbol{\theta}))$ with respect to $\boldsymbol{\theta}$, and $\tilde{g}(\boldsymbol{\theta}) = \frac{1}{|\mathcal{B}|}\sum_{i\in\mathcal{B}} \nabla_{\boldsymbol{\theta}}f(\boldsymbol{\theta}, \tilde{\mathbf{x}}_i(\boldsymbol{\theta})$ be the approximate stochastic gradient of $L_S(\boldsymbol{\theta})$. Before we prove the convergence of the algorithm, we have following assumptions.

**Assumption 1.** *The function $f(\boldsymbol{\theta}, \mathbf{x})$ satisfies the gradient Lipschitz conditions:*

$$sup_{\mathbf{x}} \|\nabla_{\boldsymbol{\theta}}f(\boldsymbol{\theta}, \mathbf{x}) - \nabla_{\boldsymbol{\theta}}f(\boldsymbol{\theta}^*, \mathbf{x})\|_2 \leq L_{\boldsymbol{\theta\theta}} \|\boldsymbol{\theta} - \boldsymbol{\theta}^*\|_2$$
$$sup_{\boldsymbol{\theta}} \|\nabla_{\boldsymbol{\theta}}f(\boldsymbol{\theta}, \mathbf{x}) - \nabla_{\boldsymbol{\theta}}f(\boldsymbol{\theta}, \mathbf{x}^*)\|_2 \leq L_{\boldsymbol{\theta}\mathbf{x}} \|\mathbf{x} - \mathbf{x}^*\|_2$$
$$sup_{\mathbf{x}} \|\nabla_{\mathbf{x}}f(\boldsymbol{\theta}, \mathbf{x}) - \nabla_{\mathbf{x}}f(\boldsymbol{\theta}^*, \mathbf{x}^*)\|_2 \leq L_{\mathbf{x}\boldsymbol{\theta}} \|\boldsymbol{\theta} - \boldsymbol{\theta}^*\|_2$$

where $L_{\boldsymbol{\theta\theta}}$, $L_{\boldsymbol{\theta}\mathbf{x}}$, and $L_{\mathbf{x}\boldsymbol{\theta}}$ are positive constants. Assumption 1 was used in (Sinha et al., 2018; Wang et al., 2019).

**Assumption 2.** *The function $f(\boldsymbol{\theta}, \mathbf{x})$ is locally $\mu$-strongly concave in $\mathcal{X} = \{\mathbf{x} : \|\mathbf{x} - \mathbf{x}_i^0\|_\infty \leq \epsilon\}$ for all $i \in [n]$, i.e., for any $\mathbf{x}_1$ , $\mathbf{x}_2 \in \mathcal{X}_i$:*

$$f(\boldsymbol{\theta}, \mathbf{x}_1) \leq f(\boldsymbol{\theta}, \mathbf{x}_2) + \langle \nabla_{\mathbf{x}}f(\boldsymbol{\theta}, \mathbf{x}_2), \mathbf{x}_1 - \mathbf{x}_2 \rangle - \frac{\mu}{2}\|\mathbf{x}_1 - \mathbf{x}_2\|_2^2$$

where $\mu$ is a positive constant which measures the curvature of the loss function. This assumption was used for analyzing distributional robust optimization problems (Sinha et al., 2018).

**Assumption 3.** *The variance of the stochastic gradient $g(\boldsymbol{\theta})$ is bounded by a constant $\sigma^2 > 0$:*

$$\mathbb{E}[\|g(\boldsymbol{\theta}) - \nabla L_S(\boldsymbol{\theta})\|_2^2] \le \sigma^2$$

*where $\nabla L_S(\boldsymbol{\theta})$ is the full gradient.*

The Assumption 3 is commonly used for analyzing stochastic gradient optimization algorithms.

**Theorem 2.** *Suppose Assumptions 1,2, and 3 holds. Denote $\Delta = L_S(\boldsymbol{\theta}^0) - \min_{\boldsymbol{\theta}} L_S(\boldsymbol{\theta})$. If the step size of outer minimization is $\eta_t = \min(1/L, \sqrt{\frac{\Delta}{TL\sigma^2}})$. Then, we have:*

$$\frac{1}{T} \sum_{t=0}^{T-1} \mathbb{E}[\|\nabla L_S(\boldsymbol{\theta}^t)\|_2^2] \le 4\sigma\sqrt{\frac{L\Delta}{T}} + 5L_{\boldsymbol{\theta}\mathbf{x}}^2\delta^2$$

*where $L = L_{\boldsymbol{\theta}\mathbf{x}}L\mathbf{x}\boldsymbol{\theta}/\mu + L_{\boldsymbol{\theta}\boldsymbol{\theta}}$.*

The proof of this theorem can be found in the Appendix. Theorem 2 indicates that if the Amata finds solutions of the inner maximization problem closer enough to the maxima, Amata can converge at a sublinear rate.

**Lemma 1.** *Under Assumption 1 and Assumption 2, we have $L_S(\boldsymbol{\theta})$ is $L$-smooth where $L = L_{\boldsymbol{\theta}\mathbf{x}}L_{\mathbf{x}\boldsymbol{\theta}}/\mu + L_{\boldsymbol{\theta}\boldsymbol{\theta}}$, i.e., for any $\boldsymbol{\theta}_1$ and $\boldsymbol{\theta}_2$ it holds:*

$$L_S(\boldsymbol{\theta}_1) \le L_S(\boldsymbol{\theta}_2) + \langle \nabla L_S(\boldsymbol{\theta}_2), \boldsymbol{\theta}_1 - \boldsymbol{\theta}_2 \rangle + \frac{L}{2}\|\boldsymbol{\theta}_1 - \boldsymbol{\theta}_2\|_2^2$$

The proof of Lemma 1 can be found in Wang et al. (2019).

**Lemma 2.** *Under Assumption 1 and Assumption 2, the norm of difference between the approximate stochastic gradient $g(\boldsymbol{\theta})$ and the stochastic gradient $\tilde{g}(\boldsymbol{\theta})$ is bounded, i.e., it holds that:*

$$\|\tilde{g}(\boldsymbol{\theta}) - g(\boldsymbol{\theta})\|_2 \le L_{\boldsymbol{\theta}\mathbf{x}}\delta$$

*Proof.* We have

$$
\begin{aligned}
\|\tilde{g}(\boldsymbol{\theta}) - g(\boldsymbol{\theta})\|_2 &= \left\| \frac{1}{|\mathcal{B}|} \sum_{i \in \mathcal{B}} \left( \nabla_{\boldsymbol{\theta}} f(\boldsymbol{\theta}, \tilde{\mathbf{x}}_i(\boldsymbol{\theta})) - \nabla \bar{f}_i(\boldsymbol{\theta}) \right) \right\|_2 \\
&\le \frac{1}{|\mathcal{B}|} \sum_{i \in \mathcal{B}} \|\nabla_{\boldsymbol{\theta}} f(\boldsymbol{\theta}, \tilde{\mathbf{x}}_i(\boldsymbol{\theta})) - \nabla_{\boldsymbol{\theta}} f(\boldsymbol{\theta}, \mathbf{x}_i^*(\boldsymbol{\theta}))\|_2 \\
&\le \frac{1}{|\mathcal{B}|} \sum_{i \in \mathcal{B}} L_{\boldsymbol{\theta}\mathbf{x}} \|\tilde{\mathbf{x}}_i(\boldsymbol{\theta}) - \mathbf{x}_i^*(\boldsymbol{\theta})\|_2
\end{aligned}
\tag{19}
$$

where the first inequality is from the triangle inequality, and the second inequality is from Assumption 1. Next, we insert Equation 18 into the above inequality then:

$$\|\tilde{g}(\boldsymbol{\theta}) - g(\boldsymbol{\theta})\|_2 \le L_{\boldsymbol{\theta}\mathbf{x}}\delta \tag{20}$$

which completes the proof. □

Now we can prove the Theorem 2:

*Proof.* From Lemma 1, we have:

$$L_S(\boldsymbol{\theta}^{t+1}) \leq L_S(\boldsymbol{\theta}^t) + \langle \nabla L_S(\boldsymbol{\theta}^t, \boldsymbol{\theta}^{t+1} - \boldsymbol{\theta}^t \rangle + \frac{L}{2} \left\| \boldsymbol{\theta}^{t+1} - \boldsymbol{\theta}^t \right\|_2^2$$

$$= L_S(\boldsymbol{\theta}^t) + \eta_t \langle \nabla L_S(\boldsymbol{\theta}^t), \nabla L_S(\boldsymbol{\theta}^t - \tilde{g}(\boldsymbol{\theta}^t)) \rangle - \eta_t \left\| \nabla L_S(\boldsymbol{\theta}^t) \right\|_2^2$$
$$+ \frac{L\eta_t^2}{2} \left\| \tilde{g}(\boldsymbol{\theta}^2) \right\|_2^2$$

$$= L_S(\boldsymbol{\theta}^t) - \eta_t(1 - \frac{L\eta_t}{2}) \left\| \nabla L_S(\boldsymbol{\theta}^t) \right\|_2^2 + \eta_t(1 - \frac{L\eta_t}{2}) \cdot$$

$$\langle \nabla L_S(\boldsymbol{\theta}^t), \nabla L_S(\boldsymbol{\theta}^t) - \tilde{g}(\boldsymbol{\theta}^t) \rangle + \frac{L\eta_t^2}{2} \left\| \tilde{g}(\boldsymbol{\theta}^t) - \nabla L_S(\boldsymbol{\theta}^t) \right\|_2^2$$

$$= L_S(\boldsymbol{\theta}^t) - \eta_t(1 - \frac{L\eta_t}{2}) \left\| \nabla L_S(\boldsymbol{\theta}^t) \right\|_2^2 + \eta_t(1 - \frac{L\eta_t}{2}) \cdot$$

$$\langle \nabla L_S(\boldsymbol{\theta}^t), \nabla L_S(\boldsymbol{\theta}^t) - g(\boldsymbol{\theta}^t) \rangle + \eta_t(1 - \frac{L\eta_t}{2}) \langle \nabla L_S(\boldsymbol{\theta}^t),$$

$$\nabla L_S(\boldsymbol{\theta}^t) - g(\boldsymbol{\theta}^t) \rangle + \frac{L\eta_t^2}{2} \left\| \tilde{g}(\boldsymbol{\theta}^t) - g(\boldsymbol{\theta}^t) + g(\boldsymbol{\theta}^t) - \nabla L_S(\boldsymbol{\theta}^t) \right\|_2^2$$

$$\leq L_S(\boldsymbol{\theta}^t) - \frac{\eta_t}{2}(1 - \frac{L\eta_t}{2}) \left\| \nabla L_S(\boldsymbol{\theta}^t) \right\|_2^2 + \eta_t(1 - \frac{L\eta_t}{2}) \cdot$$

$$\left\| \tilde{g}(\boldsymbol{\theta}) - g(\boldsymbol{\theta}^t) \right\|_2^2 + L\eta_t^2(\left\| \tilde{g}(\boldsymbol{\theta}^t) - g(\boldsymbol{\theta}^t) \right\|_2^2 + \|g(\boldsymbol{\theta}^t)$$

$$- \nabla L_S(\boldsymbol{\theta}^t)\|_2^2) + \eta_t(1 + \frac{L\eta_t}{2}) \langle \nabla L_S(\boldsymbol{\theta}^t), \nabla L_S(\boldsymbol{\theta}^t) - g(\boldsymbol{\theta}^t) \rangle$$

Note that $\mathbb{E}[g(\boldsymbol{\theta}^t)] = \nabla L_S(\boldsymbol{\theta})$, taking expectation on both sides of the inequality conditioned on $\boldsymbol{\theta}^t$. Then we use Assumption 3 and Lemma 2 and simplify the above inequality:

$$\mathbb{E}[L_S(\boldsymbol{\theta}^{t+1}) - L_S(\boldsymbol{\theta}^t)|\boldsymbol{\theta}^t] \leq -\frac{\eta_t}{2}(1 - \frac{L\eta_t}{2}) \left\| \nabla L_S(\boldsymbol{\theta}^t) \right\|_2^2$$
$$+ \frac{\eta_t}{2}(1 + \frac{3L\eta_t}{2})L_{\boldsymbol{\theta}\mathbf{x}}^2\delta^2 + L\eta_t^2\sigma^2$$

Taking the telescope sum of the above equation from $t = 0$ to $t = T - 1$, we have

$$\sum_{t=0}^{T-1} \frac{\eta_t}{2}(1 - L\frac{\eta_t}{2})\mathbb{E}[\left\| \nabla L_S(\boldsymbol{\theta}^t) \right\|_2^2] \leq \mathbb{E}[L_S(\boldsymbol{\theta}^0 - \boldsymbol{\theta}^T)]$$
$$+ \sum_{t=0}^{T-1} \frac{\eta_t}{2}(1 + \frac{3L\eta_t}{2})L_{\boldsymbol{\theta}\mathbf{x}}^2\delta^2 + L\eta_t^2\sigma^2$$

We set $\eta_t = \min(1/L, \sqrt{\frac{\Delta}{TL\sigma^2}})$, we have

$$\frac{1}{T}\sum_{t=0}^{T-1}\mathbb{E}[\left\| \nabla L_S(\boldsymbol{\theta}^t) \right\|_2^2] \leq 4\sigma\sqrt{\frac{L\Delta}{T}} + 5L_{\boldsymbol{\theta}\mathbf{x}}^2\delta^2$$

Thus, we complete the proof. □

## D    APPENDIX: APPROXIMATION OF THE OPTIMAL CONTROL CRITERION

In this section we show how to derive the approximation (14) from (13). The primary assumption we make is that $T_2 - T_1 = \eta \ll 1$, which allows one to use a local approximation for various functions to derive a simple-to-compute criterion. For this reason, the derivation here will be largely heuristic.

First, we assume that we consider a constant control $\mathbf{u}_s \equiv \mathbf{u}$ on the interval $s \in [T_1, T_2] \equiv [t, t+\eta]$. Next, recall that the adversarial robustness which serves as our terminal loss function for the control problem is $\Phi(\boldsymbol{\theta}) = \max_{\mathbf{z}} \ell(h_{\boldsymbol{\theta}}[\mathcal{A}_{\boldsymbol{\theta},\mathbf{z}}(x)], y)$. In practice, any control that sufficiently conducts the

adversarial perturbation serves as a good approximation. This is because the DNNs are very prone to adversarial attacks before the convergence of adversarial training and the real-time adversarial loss is close to the worst-case adversarial loss. Thus, we can approximate $\Phi(\boldsymbol{\theta})$ by $\ell(h_{\boldsymbol{\theta}}[\mathcal{A}_{\boldsymbol{\theta},\mathbf{z}}(x)], y)$ by some chosen $\mathbf{z}$ that makes practical computations easy.

Now, Assuming regularity we can expand the state equation (10) to get to leading order as $\eta \to 0$

$$\boldsymbol{\theta}_s^{\mathbf{u}} = \boldsymbol{\theta}_t + o(1), \qquad s \in [t, t+\eta]. \tag{21}$$

On the other hand, the co-state equation (11) gives

$$\mathbf{p}_s^{\mathbf{u}} = -\nabla_{\boldsymbol{\theta}}\Phi(\boldsymbol{\theta}_{T_2}) + o(1) = -\nabla_{\boldsymbol{\theta}}\ell(h_{\boldsymbol{\theta}_t}[\mathcal{A}_{\boldsymbol{\theta}_t,\mathbf{z}}(x)], y) + o(1), \qquad s \in [t, t+\eta]. \tag{22}$$

But, by definition $F(\boldsymbol{\theta}, \mathbf{u}) = -\nabla_{\boldsymbol{\theta}}\ell(h_{\boldsymbol{\theta}}[\mathcal{A}_{\boldsymbol{\theta},\mathbf{u}}(x)], y)$. Then, for any $s \in [t, t+\eta]$, we have from (21) and (22) that

$$\begin{aligned} H(\boldsymbol{\theta}_s^{\mathbf{u}}, \mathbf{p}_s^{\mathbf{u}}, \mathbf{v}) &= \langle \mathbf{p}_s^{\mathbf{u}}, F(\boldsymbol{\theta}_s^{\mathbf{u}}, \mathbf{v}) \rangle - R(\mathbf{v}) \\ &= \langle \nabla_{\boldsymbol{\theta}}\ell(h_{\boldsymbol{\theta}_t}[\mathcal{A}_{\boldsymbol{\theta}_t,\mathbf{z}}(x)], y), \nabla_{\boldsymbol{\theta}}\ell(h_{\boldsymbol{\theta}_t}[\mathcal{A}_{\boldsymbol{\theta}_t,\mathbf{v}}(x)], y), \rangle - R(\mathbf{v}) + o(1) \end{aligned} \tag{23}$$

Note that the point of which the various expressions the above are expanded from appears arbitrarily chosen from a strict mathematical sense, but these choices enable fast practical computation. Any of such expansions, assuming enough regularity, are equivalent in the limit $\eta \to 0$.

Plugging the above into the expression for $C$ (Eq. 13), we have

$$\begin{aligned} C(\mathbf{u}, t) \approx \max_{\mathbf{v}} \, & \{ \langle \nabla_{\boldsymbol{\theta}}\ell(h_{\boldsymbol{\theta}_t}[\mathcal{A}_{\boldsymbol{\theta}_t,\mathbf{v}}(x)], y), \nabla_{\boldsymbol{\theta}}\ell(h_{\boldsymbol{\theta}_t}[\mathcal{A}_{\boldsymbol{\theta}_t,\mathbf{z}}(x)], y) \rangle - R(\mathbf{v}) \} \\ & - (\langle \nabla_{\boldsymbol{\theta}}\ell(h_{\boldsymbol{\theta}_t}[\mathcal{A}_{\boldsymbol{\theta}_t,\mathbf{u}}(x)], y), \nabla_{\boldsymbol{\theta}}\ell(h_{\boldsymbol{\theta}_t}[\mathcal{A}_{\boldsymbol{\theta}_t,\mathbf{z}}(x)], y) \rangle - R(\mathbf{u})) \end{aligned} \tag{24}$$

As discussed in the beginning of this section, to ease computation, we may replace $\mathbf{z}$ by $\mathbf{v}$ in the first term and $\mathbf{u}$ in the second term, assuming that they give sufficient adversarial perturbations to get

$$\begin{aligned} C(\mathbf{u}, t) \approx \max_{\mathbf{v}} \, & \{ \|\nabla_{\boldsymbol{\theta}}\ell(h_{\boldsymbol{\theta}_t}[\mathcal{A}_{\boldsymbol{\theta}_t,\mathbf{v}}(x)], y)\|^2 - R(\mathbf{v}) \} \\ & - (\|\nabla_{\boldsymbol{\theta}}\ell(h_{\boldsymbol{\theta}_t}[\mathcal{A}_{\boldsymbol{\theta}_t,\mathbf{u}}(x)], y)\|^2 - R(\mathbf{u})) . \end{aligned} \tag{25}$$

Upon substituting $\mathbf{v} = (K, \alpha)$ and $\mathbf{u} = (K^t, \alpha_t)$ gives the desired result.

