# OpenReview forum: "Amata: An Annealing Mechanism for Adversarial Training Acceleration"
_ICLR.cc/2020/Conference — Reject_

### Official Review · AnonReviewer3 · 2019-10-24
**Official Blind Review #3**

**Rating:** 6

**Review:**

This paper proposed an annealing mechanism for PGD adversarial training in Madry et al. This mechanism gradually reduces the step size and increases the number of iterations of PGD maximization. The authors claim that the proposed method can achieve comparable performance as the original PGD adversarial training with less time.

First I think this mechanism itself's contribution is kind of incremental because this technique is well under the PGD adversarial training framework. In fact, there are existing works trying to accelerate adversarial training, for example [1]. I think the authors may need to compare their proposed method with [1] if possible.

I like the optimal control formulation in 2.2 and 2.3. More discussion on (8) would be welcomed. In particularly, I think the authors implicitly define the training cost as the sum of iterations in all PGD attacks and turn it into an integral. More explanation here would help. Also, it seems that the reason why \alpha=\frac{\tau}{K^t} is not fully explained in the paper. The authors choose to fix the total step length of PGD. Is this a heuristics?

In the experimental part, I think it would be better to also report the number of SGD steps on \theta to get a sense of convergence. I am not sure the acceleration is due to fewer PGD steps at each t, or fewer training epoch T.

[1] Zhang, Dinghuai, et al. "You Only Propagate Once: Accelerating Adversarial Training via Maximal Principle." NeurIPS 2019.

**Experience Assessment:**

I have published one or two papers in this area.

**Review Assessment: Checking Correctness Of Derivations And Theory:**

I assessed the sensibility of the derivations and theory.

**Review Assessment: Checking Correctness Of Experiments:**

I assessed the sensibility of the experiments.

**Review Assessment: Thoroughness In Paper Reading:**

I read the paper at least twice and used my best judgement in assessing the paper.

---

> ### Author Response · Authors · 2019-11-15
> **Rebuttal to reviewer 3**
>
> Thank you very much for your positive comments.
>
> On clarity in optimal control formulation:
> We have improved clarity by adding more explanation to the terms and the results. In particular, we added some explanation of the form of the optimization problem presented in (8).
> Cost as integral: Yes. In our specific application here the cost is the total cost due to PGD iterations. However, the formulation there is general: R(u) is the cost per unit time introduced by applying a hyper-parameter selection u.
> why \alpha=\frac{\tau}{K^t}: This is to restrict the control space G to be a one-dimensional space. In this sense it is a heuristic restriction so as to save computation.
>
> On Yopo:
> The goal of this paper is to have a practically useful method to accelerate adversarial training. We use the example of controlling the PGD adversarial training because the popularity and simplicity of implementing PGD adversarial training. However, our method is definitely not limited within the framework of PGD. As stated in our paper, we can potentially combine our method with other adversarial training methods,  such as YOPO, the results are shown in Appendix B.3. We also reported the convergence with regard to the number of epochs in Appendix B.2. We found that the storage space saving is mainly due to the savings in the number of PGD steps as expected. The intuitive explanation of this phenomenon is that if the outer loop converges faster, it indicates that the adversarial examples generated during the training process are too weak.

---

### Official Review · AnonReviewer2 · 2019-10-24
**Official Blind Review #2**

**Rating:** 6

**Review:**

Summary: Deep learning algorithms are known to be prone to adversarial examples. This paper proposes a simple modification for adversarial training in order to improve the robustness of the algorithms. The adversarial training of the neural networks involve two steps: the outer loop where the loss is being minimized, the inner loop where the adversarial examples are being found by using PGD. The inner PGD optimization makes training more expensive, because for every gradient step, algorithm needs to perform several steps of PGD. This paper proposes a simple modification to the PGD that is being used in the inner loop. The proposed modification involves the increasing the number of adversarial training steps and decreasing the adversarial training step size gradually as the training proceeds. Then, they analyze the modifications from the theory of optimal control. Their experiments show that their method can achieve similar robustness to the other existing approaches with less computations.

Pros:
 - Interesting topic, important subject and a very simple approach.
 - Significant improvements in terms of training times.

Cons:
- The math notation is a bit cumbersome, lots of undefined variables and functions used without properly giving enough background to the readers. This field is still new not every reader might be familiar with the optimal control theory or the common notation that is being used there. Please try to explain the equations and theorems more carefully.
- Experiments are only on small-scale toy-datasets like CIFAR10 and MNIST.

Complete Assessment: I like the proposed approach, in this paper. It is a fairly simple approach and seems to work well in practice, at least on the tasks that the authors have tried. The writing, I think still needs some work, some of the math notation is mostly not properly explained, in Sections 2.2 and 2.3. Overall, I am not sure how much those two sections are actually contributing to the paper.

Question: Did you try using a different annealing mechanisms, besides linear decaying, such as an exponential one?

**Experience Assessment:**

I have read many papers in this area.

**Review Assessment: Checking Correctness Of Derivations And Theory:**

I assessed the sensibility of the derivations and theory.

**Review Assessment: Checking Correctness Of Experiments:**

I assessed the sensibility of the experiments.

**Review Assessment: Thoroughness In Paper Reading:**

I read the paper at least twice and used my best judgement in assessing the paper.

---

> ### Author Response · Authors · 2019-11-15
> **Rebuttal to reviewer 2**
>
> Thank you very much for your comments.
>
> On the clarity role of optimal control:
> In our revision, we have improved the clarity by adding explanations of the different variables and the optimal control results used (e.g. the optimal control criterion). See also the answers to clarity questions to reviewer 1.
> The role of optimal control in this paper is two-fold. First, it can be seen as a general formulation, as we develop in Section 2.2, of adversarial training with hyper-parameters in the inner loop. This is not limited to PGD. Second, we use approximate criterion (Eq. 13 and 14) derived from the maximum principle in optimal control to quantify the approximate optimality of adversarial training algorithms, including the one proposed in this paper.
>
> On other annealing(decay) schedules:
> Finding better annealing(decay) scheme is an interesting future direction. However, using more complex schemes may introduce extra parameters, while one of the major challenges for machine learning methods to be practically useful for industry practitioners is the complexity of the method. We tried another exponential scheme K _{t}=K_{min} +  (K_{max}-K_{min})* (1-exp^{-eta * t})/(1-exp^{-eta*T}), where K_{t} is the number of PGD adversarial training steps at t-th epoch, K_{min} is the minimum number of PGD steps in the beginning, K_{max} is the maximum number of PGD steps in the last epoch, eta is the hyper-parameter controlling the shape of the scheme. This design ensures that when t=0 K_{t} = K_{min}, when t = T, K_{t} = K_{max} to make it comparable with the linear scheme used in our paper. However, we tried different parameters of eta ranging from  0.1 to 5, but did not find performance improvements. The results are shown in Appendix B.5. We will try experiments on the large-scale datasets, such as ImageNet, in our future work because of our current limited computational facility.

---

### Official Review · AnonReviewer1 · 2019-11-02
**Official Blind Review #1**

**Rating:** 3

**Review:**

The submission follows a recent line of work that formulates adversarial training (https://arxiv.org/abs/1412.6572) as a differentiable game and attempts to reduce the computational complexity of solving the inner maximization, corresponding to finding (an) adversarial example(s) for a given parameter setting. The authors propose a way to anneal the truncation of the inner iteration, and investigate a quantity to represent the suboptimality of the truncation.

Strengths:
- (From Table 3), the method is competitive with adversarial training (PGD; Madry et al. (2017)) as well as an improved method for adversarial robustness (FOSC; Wang et al. (2019)), while requiring half the computation time.
- The writing is very clear.

Weaknesses:
- Significance: The technique AMATA itself, is a  heuristic to gradually increase the length of the inner loop optimization in adversarial training. It is unrelated to the optimal control formulation.
- Novelty: Much of the motivation from the perspective of an optimal control formulation of adversarial training is similar to https://arxiv.org/abs/1803.01299, including the PMP and successive approximation components; Eq. (13) of the submission is the continuous version of Theorem 2 in this paper.
- Some experimental details are missing.
- Minor clarity issues (see below).

Questions for the authors:
- Could the authors better contrast their work in reference to prior work (https://arxiv.org/abs/1803.01299)?
- Could the authors clarify if they use random restarts for the PGD attacks? How was the number of iterations for the PGD attack (40) selected? Why was the iteration count of the PGD attack not varied?

Clarity issues:
- Please provide references for the optimal control formulation in 2.2.
- Some references to variables in the text are not made precise by identifying the variable in question (e.g.: "a set of optimal parameter choices"; "requires information on the entire training interval").
- The function of the hyperparameters \alpha, \gamma, K are not well discussed in Section 2.4.
- Reduce plot sizes and include more experimental details in the main text.

**Experience Assessment:**

I have read many papers in this area.

**Review Assessment: Checking Correctness Of Derivations And Theory:**

I assessed the sensibility of the derivations and theory.

**Review Assessment: Checking Correctness Of Experiments:**

I assessed the sensibility of the experiments.

**Review Assessment: Thoroughness In Paper Reading:**

I read the paper thoroughly.

---

> ### Author Response · Authors · 2019-11-15
> **Rebuttal to reviewer 1**
>
> The goal of this paper is to have a practically useful method to accelerate adversarial training with numerical justification by the optimal control formulation. Because computing Hamiltonian in the optimal control formulation itself will cost a lot of time thus hindering the acceleration.  However, our formulation clearly provides a principled way for designing fast adversarial training method that is open for future research.
>
>
> Contrasting with [LH18] (https://arxiv.org/abs/1803.01299)
> Although both optimal control methodologies are used in this work and [LH18], they are fundamentally very different settings:
> Time dimension and state for the dynamics: [LH18] studies dynamics of transformation of hidden states in deep learning, and the role of the  time dimension is played by the layers. The states are the hidden activations in a deep neural network. In this work, the time dimension is the outer-loop (SGD)  iterations in adversarial training, and the states which are evolving in time are the weights of the neural network as training proceeds.
> Control variables: [LH18] basically generalizes the back-propagation algorithm for training neural networks through the PMP. The control variables are the weights in the neural network. In this work, the control variables are the hyper-parameters for adversarial training.
> In other words, the only similarity between this work and [LH18] is that both use optimal control ideas (but the same can be said about any work that uses optimal control or related techniques in calculus of variations). We have clarified this point briefly in the revised manuscript after the reference of  [LH18] on the MSA algorithm.
>
>
> Other questions:
> PGD-40 is a suitably strong attack for evaluation that is used in other papers. We also tried other numbers of iterations, such as PGD-20 and PGD-60.  The results are similar and are shown in Appendix B.4.
>
> Random restarts were used for PGD.  It seems that random starts will not have large effects on the results (https://github.com/MadryLab/mnist_challenge/issues/6).
>
> Clarity issues raised:
> “-Significance: The technique AMATA itself, is a heuristic to gradually increase the length of the inner loop optimization in adversarial training. It is unrelated to the optimal control formulation.”
> As commented by Reviewer 2, The proposed modification is analyzed by the optimal control theory to check its optimality, quantified by the derived . The optimal control analysis and numerical experiments demonstrate that though simple, the proposed mechanism is effective for accelerating adversarial training without sacrificing the robustness.
>
> “- Please provide references for the optimal control formulation in 2.2.”
> The formulation in 2.2 is part of the original work in this paper. This is made clear in contribution 2 in the introduction. The control theoretic tools such as the Pontryagin’s maximum principles have already been cited.
> “- Some references to variables in the text are not made precise by identifying the variable in question (e.g.: "a set of optimal parameter choices"; "requires information on the entire training interval").”
> in our specific application, these are the optimal choices of inner-loop hyper-parameters as the training proceeds. We have added this clarification in the revision.
> “- The function of the hyperparameters \alpha, \gamma, K are not well discussed in Section 2.4.”
> \alpha is the step-size, \gamma is the computational cost of each PGD step and K is the number of PGD steps per outer iteration. These information are shown in Alg. 1, but for clarity we added additional explanation of these in section 2.4.

---

### Author Response · Authors · 2019-11-15
**Overall rebuttal**

We appreciate all reviewers for providing insightful comments on technical aspect of the proposed method. In this paper, we have provided a simple yet effective mechanism to accelerate the adversarial training with only a few lines of code. The optimality of the proposed method is numerically justified with a criterion based on the optimal control theory.

Below, we summarize the revision briefly. Detailed responses to each reviewer/comment are followed based on each reviewer feedback.
1.Clarification on novelty:
Although both optimal control methodologies are used in this work and [LH18], the formulation of the optimal control problem is fundamentally different. The latter is interested in developing a control viewpoint of deep neural networks idealized as a dynamical system, where as the current work formulates the adversarial training problem as a control problem.
The only overlap is the mathematical tools used. Details are given below, and we have also added some clarifications in the paper on this point.
2.Clarification on the importance of optimal control formulation:
The proposed modification is analyzed by optimal control theory to quantify its deviation  from optimality through the derived criterion C, which is shown to be small for our method. The optimal control analysis and numerical experiments demonstrate that though simple, the proposed mechanism is effective for accelerating adversarial training without sacrificing the robustness.
3.Combination of other acceleration methods:
We show an example combining the Amata and YOPO in the Appendix B.3. The additional example shows that our method can be easily incorporated within other methods, such as YOPO, to further reduce almost 50 percent of the computation time and achieve even better performance when converged.
4.Other annealing(decay) schemes, such as exponential scheme:
We added additional experiments on other annealing(decay) schemes, such as an exponential scheme. The results do not out-perform the current scheme. We do not claim that the current simple scheme is optimal, but to develop more complex schemes one would probably have to appeal to more sophisticated tools, such as adaptive or control based methods. Our current approach emphasizes simplicity, and we show both in terms of the optimal control criterion and experiments that they perform adequately well.

---

### Decision · Program_Chairs · 2019-12-19

**Decision:**

Reject

**Comment:**

The paper proposes a modification for adversarial training in order to improve the robustness of the algorithm by developing an annealing mechanism for PGD adversarial training. This mechanism gradually reduces the step size and increases the number of iterations of PGD maximization. One reviewer found the paper to be clear and competitive with existing work, but raised concerns of novelty and significance. Another reviewer noted the significant improvements in training times but had concerns about small scale datasets. The final reviewer liked the optimal control formulation, and requested further details. The authors provided detailed answers and responses to the reviews, although some of these concerns remain. The paper has improved over the course of the review, but due to a large number of stronger papers, was not accepted at this time.